# On the Plant Developmental Unit: From Virtual Concept to Visual Plantlet

**DOI:** 10.3390/plants14030396

**Published:** 2025-01-28

**Authors:** Shu-Nong Bai

**Affiliations:** School of Life Sciences, Peking University, Beijing 100871, China; shunongb@pku.edu.cn

**Keywords:** plant developmental unit (PDU), *Wolffia*, virtual embryo, plantlet model, coral-like colony

## Abstract

This study introduces the concept of the plant developmental unit (PDU) and validates its application using *Wolffia* Horkel ex Schleid (Araceae) as a model system for exploring fundamental processes in plant morphogenesis. Revisiting long-standing contradictions in plant biology, the author proposes viewing plants as coral-like colonies composed of multiple developmental units rather than as unitary-animal-like organisms. Utilizing the “Plant-on-Chip” culture platform, the research demonstrates *Wolffia*’s minimalist structure as a powerful model for investigating core regulatory mechanisms of plant development. The study emphasizes the pivotal role of “induction” in morphogenetic processes and highlights *Wolffia*’s potential to facilitate a paradigm shift in plant developmental biology while unlocking its applications in a second agricultural revolution. This work underscores *Wolffia*’s value in bridging fundamental research and innovative agricultural solutions.

## 1. Introduction

Plants do not speak. They do not need to tell other creatures who they are and how they live. Having thrived on Earth for over 400 million years, long before humans emerged some 200,000 to 300,000 years ago, plants exist indifferent to how we perceive or describe them.

Humans, on the other hand, have historically paid little attention to what plants are or how they live, focusing primarily on which parts are edible, as well as other utilities for millennia. It is only in recent centuries that we began collecting plants from around the world, classifying them, and attempting to explain their nature and life processes—explanations shared among humans, not with plants.

Conventionally, the study of plants is often driven by two primary objectives: satisfying human curiosity and improving the utilization of plants as living resources. If we consider that the uniqueness of humans lies in their cognitive ability—a symbolic medium that integrates essential components for sustaining a living system [1]—these two objectives ultimately converge into one: ensuring that the symbols created and used to describe plants align with the characteristics they aim to represent. This convergence highlights two fundamental requirements. First, the symbols must accurately match the features of the described objects; otherwise, they become meaningless. Second, the relationships between symbols must be self-consistent; without this, effective communication is impossible, leading to wasted energy in processing incoherent or nonsensical information.

As a plant biologist emeritus, I often ask myself: If plants could evaluate my performance as their spokesman—though they never hired me—how would they rate me? I wonder if other colleagues share this sentiment. This feeling stems from my professional experience, where I have encountered numerous self-contradictory notions that continue to circulate without rigorous scrutiny. These contradictions have been a source of frustration for me. As someone who aspires to be a qualified spokesperson for plants, I feel a responsibility to confront and address these issues, even if I cannot entirely resolve them.

## 2. Examples of Self-Contradictory Notions in Plant Biology

### 2.1. Is a Flower a Reproductive Organ or a Compressed Shoot?

The first contradictory notion I encountered in my career was regarding the definition of a flower.

Flowers have existed for more than a hundred million years before the emergence of humans. During my first biology course, Botany, in early 1978, my college lecturer presented two seemingly contradictory statements. In one instance, he taught that plants have six types of organs: roots, stems, leaves, flowers, fruits, and seeds. However, in another instance, he described a flower as a compressed shoot. Since a shoot is defined as a structure comprising a stem and leaves, and a flower consists of four parts (sepals, petals, stamens, and carpels), each derived from leaves, I was perplexed: how can a flower, composed of several organs, be categorized as a single organ?

Thirty-five years later, I discovered by chance that this self-contradictory explanation still persists in various versions of botany textbooks in China. This sparked my curiosity: How did such a problem arise? Were people aware of it, and why had it remained unresolved for so many years? After extensively reviewing classic botany textbooks and consulting with several experts, I came to understand that the issue originated from interruptions in professional training during periods of social upheaval in China, coinciding with the adoption of contemporary botany concepts from abroad.

While mainstream English-language textbooks in plant biology generally define a flower as a compressed shoot, research articles often treat flowers as a distinct unit comparable to roots, shoots, and leaves. This inconsistent usage introduces significant and unnecessary confusion, particularly when attempting to study organ-specific regulatory mechanisms.

### 2.2. Is the Plant Developmental Program Indeterminate or Determinate?

About ten years after my undergraduate botany course, I encountered another contradictory notion during my PhD studies.

While reading a review article [2], I came across the idea that the plant developmental program is characterized as “indeterminate”. This immediately reminded me of what I had learned in my undergraduate plant physiology course, where plants were described as having “unlimited growth”. However, upon further reflection, I realized a critical problem: if a developmental program is truly indeterminate, how does the next generation arise?

In any standard undergraduate botany course, students learn about the alternation of generations [3]. This theory describes a plant’s life cycle as consisting of a sporophytic generation and a gametophytic generation. The sporophytic generation is a diploid multicellular structure that begins with a zygote and culminates in spore production. Conversely, the gametophytic generation is a haploid multicellular structure that arises from a spore and ends with gamete production. These gametes then fuse to form a zygote, initiating a new life cycle.

If we accept the alternation of generations as a fundamental framework, the concept of an “indeterminate” developmental program or “unlimited growth” seems inherently contradictory. How can a cycle, by definition determinate, coexist with a notion of endless growth?

After more than twelve years’ study on cucumber unisexual flower development, I realized that the core process underlying the alternation of generations is what I called the “sexual reproduction cycle (SRC)” [4]. The SRC emerges along with the emergence of eukaryotes and occurs at the unicellular level, consisting of a diploid zygote, a meiotic cell, which would commit to meiosis, and four haploid gametes, which would pair and fuse to form two new diploid zygotes of next generation. Since the net outcome, in terms of change of cell numbers, is one diploid zygote becomes two diploid zygotes during the process, which is similar to one round of the mitotic cell cycle, it was designated as the sexual reproduction cycle.

From the perspective of the SRC, it is clear that determinacy mainly refers to the genetic aspect of the life cycle. After multicellularization occurred there emerges an issue of relationship between the multicellular structures and the unicellular SRC. Then, there emerges the issue of whether morphogenetic processes are determinate or indeterminate, respectively in relation to animal and plant bodies. If we think deeper to distinguish the existing levels of the objects we are talking about, the contradiction might vanish.

### 2.3. Is the Default State in the Plant Life Cycle Reproductive or Vegetative?

Most plant biologists focus on angiosperms as experimental materials, primarily because they include many of the world’s most important crops. In angiosperms, sporophyte development is dominant and is generally divided into two phases: the vegetative phase, which increases plant size, and the reproductive phase, which leads to spore production and gametophyte development. Since the sporophyte originates from a zygote and initially produces photosynthetic organs, the conventional belief is that vegetative development occurs first and represents the default state.

However, during my postdoctoral research with Renee Sung at UC Berkeley, I worked on the *emf* mutant project. The *emf* mutant is a single recessive mutant that flowers immediately after seed germination without forming rosettes [5,6]. As recessive mutations typically result in the loss of gene function, the *emf* phenotype indicates that the gene required for vegetative growth is absent. Since reproductive growth (i.e., flowering) remains intact in the *emf* mutant, this suggests that flowering does not depend on prior vegetative growth. Instead, it implies that reproductive development might be the default state, contradicting the conventional view.

Through Renee, I learned that this idea was not new. In the early 20th century, British botanist F. O. Bower proposed that reproductive development is the default state, with vegetative development interpolated between the zygote and meiotic cells [7]. However, Bower’s theory, based on his work on pteridophytes, was largely overlooked, as pteridophytes have been marginalized in mainstream plant biology. Without my experience with the *emf* mutant project, I might never have encountered Bower’s insightful theory.

### 2.4. Is a Plant a Unitary-Animal-like Individual or a Coral-like Colony?

If the plant life cycle or developmental program is determinate—starting from a zygote and ending with gametes that fuse to form the next generation’s zygote—a puzzling phenomenon arises: How can a tree simultaneously bear fruits containing next-generation seeds while producing vegetative shoots? Has the tree completed its life cycle, or has it not?

During my time at Berkeley, I sought reference points to compare the life cycles of all land plants. Through reviewing various botany textbooks, I identified three universal events at the unicellular level in plant life cycles: meiosis, gametogenesis, and fertilization. Using meiosis and fertilization as reference points, I integrated the life cycles of bryophytes, pteridophytes, and spermatophytes into a unified diagram (Figure 1A). To better represent the complexity changes in gametophytes, I later adapted the diagram into a sine-wave shape in the late 1990s (Figure 1B). This comparison revealed a clear trend: from bryophytes to pteridophytes to spermatophytes, organ types become more complex in sporophytes and more simplified in gametophytes, aligning with Hofmeister’s “alternation of generations” theory.

One intriguing insight emerged from Figure 1: while the number of organs in plants can be unlimited, the types of organs are relatively fixed. For instance, in *Arabidopsis*, the sporophyte completes its life cycle with seven types of lateral organs—cotyledons, rosette leaves, cauline leaves, sepals, petals, stamens, and carpels—developed between the zygote and meiosis. Considering that gametophyte development is canalized within spores, the life cycle of *Arabidopsis* can be seen as the sequential completion of organ types. If we simplify this concept for visual clarity, reducing the number of organs to two, the resulting structure (Figure 2A) could represent the basic unit of life cycle completion. This unit was designated as the plant developmental unit (PDU) [8,9].

Returning to the initial puzzle, it is evident that all plants, whether annual or perennial, grasses or trees, complete their life cycle. To resolve the paradox, we must shift our perspective: instead of treating a plant as a unitary-animal-like individual, we should view it as a coral-like colony composed of multiple developmental units (PDUs) at various stages of development.

This shift initially left me in self-doubt, as it diverged significantly from mainstream botanical frameworks. This uncertainty persisted for years until Prusinkiewicz visited Beijing in 2013. During his visit, he introduced not only his elegant work in algorithmic botany [10] but also Agnes Arber’s *The Natural Philosophy of Plant Form* [11]. Through Arber’s work and other historical references [12], I discovered that the concept of a plant as a colony, with buds as unit of life cycle completion, was originally proposed by the founding fathers of modern botany in the 17th century, and further developed in the early 19th century [13,14]. My proposal of the PDU concept, I realized, was merely a reinvention of this historical idea due to my ignorance of earlier literature.

Further reading revealed reasons for the shift from a bud-centered view to a plant-centered view. While this divergence offered certain conveniences for experimental design, such as sample collection and measurements, it created significant challenges for detailed analyses and logical interpretations by obscuring the modular nature of plant development. Understanding plants as colonies of PDUs restores clarity and precision to such studies, aligning with their fundamental biological reality.

### 2.5. Is a Plant Body Derived from a Foliage or Axial Structure?

The innovative work in algorithmic botany by Prusinkiewicz and Lindenmayer [10] addresses another longstanding debate: whether a plant body is fundamentally derived from axial or foliage structures. In *The Algorithmic Beauty of Plants*, they propose that a plant body is essentially an “axial tree” (Figure 3A). This concept allowed them to successfully simulate a wide variety of plant structures using computational models. Conversely, the success of their simulations also supports the validity of the “axial tree” as an abstract model for plant morphogenesis, which serves as the vehicle for life cycle completion. This model revives Zimmermann’s telome theory, first introduced in the 1930s (Figure 3B) [15,16,17].

In contrast, another influential theory suggests that a plant body consists primarily of foliage structures undergoing metamorphosis [11,18,19]. These two perspectives highlight a fundamental geometric distinction: axial structures can be abstracted as one-dimensional, while foliage structures are two-dimensional. Determining which of these models better aligns with biological reality is crucial for accurately understanding the structural evolution and development of plant bodies.

## 3. The Concept of PDU, the Rediscovery of *Wolffia*, and the Crowdfund Project

Before realizing that the concept of the plant developmental unit (PDU) was essentially a reinvention of a historical idea, I shared it with friends to gather critical feedback and suggestions. Upon my return from Berkeley in 1994, I introduced the concept to my colleague and friend Da-Ming Zhang at the Institute of Botany, CAS. He was excited and encouraged me to conduct experiments to validate the concept. To elaborate on the idea, he coined the term “virtual embryo” (represented by the yellow region in Figure 2), comparing it to an animal embryo, which is physically visible and initiates and differentiates its primordia principally in synchrony. Unlike animal embryos, however, plant “virtual embryos” initiate and differentiate their primordia sequentially, collectively serving as vehicles for life cycle completion.

When Da-Ming suggested experiments to demonstrate the PDU concept, I expressed skepticism. Plants consist of unsynchronized PDUs due to continuous branching, making such experiments challenging. Da-Ming immediately suggested *Wolffia* as a suitable model species. While I found his description of *Wolffia* intriguing, I was hesitant to explore it at that time. As a young researcher, I had to focus on areas where I had expertise, avoiding high-risk ventures into unfamiliar territory.

Unexpectedly, a few years later, Da-Ming committed to *Wolffia* research in his own lab. He collected samples across China, analyzed genetic structures, described morphogenesis, and developed laboratory cultivation methods. However, a major challenge arose: *Wolffia globosa* plantlets did not flower under his lab conditions, hindering research on its complete life cycle. In 2007, recognizing Da-Ming’s challenge, I felt obligated to help. I sought collaborations with physicists specializing in microfluidic devices to develop a culture system that might optimize conditions for *Wolffia* flowering.

From my literature search, I learned that although *Wolffia* is a marginalized angiosperm species, Swiss botanist Elias Landolt systematically studied and maintained its stocks [20,21]. While there are different views, this genus is classified under Araceae [22,23]. There are eleven species recorded in the genus. *Wolffia* represents the smallest angiosperms in the world, characterized by a round-shaped plantlet of about 1 mm in diameter. Before *Arabidopsis* became the dominant model plant, the Lemnaceae family, including *Wolffia*, was considered a potential group of model organisms [24,25,26]. However, due to technological limitations, studies on *Wolffia*’s morphogenesis and life cycle remained incomplete when I joined the research. Under a crowdfunding framework, a team of experts from diverse fields recognized the significance of the *Wolffia* study and contributed to the Plant-on-Chip project, yielding valuable insights [26].

Using advanced tools, we investigated *Wolffia*’s plantlets. From a single growth tip, leaf primordia are produced sequentially in one direction, without complicated phyllotaxy. Each plantlet contains an average of three leaf primordia, aligned from the deepest bottom of the cave outward (Figure 4A. To avoid repeat description, for detailed information about the morphology, see the figures in reference [20,21], and the Figure 1 in [26]). As primordia grow, they form their own growth tips and detach via an abscission layer. Stress treatments induce the formation of two new primordia opposite the original direction of leaf primordia initiation. Within five days, a crack appears on the surface, revealing one stamen and one pistil. Remarkably, a typical *Wolffia* plantlet comprises three “organs”—one leaf, one stamen, and one pistil—without roots, a typical stem, or vascular bundles (A detailed description is available in Figure 1 in [26]).

The Plant-on-Chip culture platform now enables the detailed tracking of individual *Wolffia* plantlets, facilitating precise analysis of morphogenetic processes. High-quality genomic, transcriptomic, and even protein structure data are available on the dedicated website www.wolffiapond.net.

Despite this progress, one issue remains unresolved: seeds have not been observed in lab-grown *Wolffia*. This may stem from challenges in pollination, fertilization, or embryogenesis, but these obstacles could be overcome with the involvement of experts in related fields. As a result of the inability to generate seeds in lab-grown *Wolffia*, little information is available regarding its developmental genetics, apart from some genomic analyses.

Nevertheless, the establishment of the Plant-on-Chip system has fulfilled a long-standing goal: using *Wolffia* as an experimental system to demonstrate the PDU concept. The simplified structure of *Wolffia* plantlets, serving as approximate PDUs, allows researchers to study core morphogenetic processes without interference from the complexity of unsynchronized PDUs in branching colonies.

## 4. Fundamental Issues in Plant Morphogenesis Revealed by the *Wolffia* Investigation and the PDU Concept

Through the investigation of *Wolffia* morphogenesis, the plant developmental unit (PDU) concept illustrated in Figure 2 has been transformed from a theoretical framework into a tangible model, represented by a visible *Wolffia* plantlet. The morphogenesis of *Wolffia* exemplifies core processes that are likely universal across all angiosperms, highlighting several fundamental issues in plant morphogenesis that have not been adequately addressed in conventional research approaches.

### 4.1. Tip Growth and Shoot Apical Meristem: Decoupling Structure and Function

A key issue in understanding plant morphogenesis is the relationship between the structure and function of the shoot apical meristem (SAM). This topic dates back to the 18th century when Wolff (1759) first described tip growth in shoots [27]. Since then, growth tips in shoots have been extensively studied [28,29,30,31]. It is now understood that while tip growth is a common phenomenon across all land plants, the structural organization of growth tips varies significantly. In bryophytes, the growth tip often consists of a single cell; in pteridophytes, it comprises a few cells; in gymnosperms, it includes a group of cells without a distinct layered structure; and in angiosperms, the growth tip features a well-defined tunica–corpus structure [32] (There are other ways of description of angiosperm SAM structures in addition of tunica–corpus theory, such as zonation and organizing center [33,34]). This structural diversity suggests that specific multicellular configurations may not be essential for the functional capabilities of tip growth in all organisms.

However, for decades, functional analyses of growth tips have largely focused on angiosperms. Researchers often identify genes required for SAM function by screening mutants with aberrant tunica–corpus structures. Genes causing defects in this structure are typically considered critical for SAM functionality [31].

The relationship between structure and function in *Wolffia*, however, challenges this widely held notion. As an angiosperm, *Wolffia* exhibits fully functional growth tips capable of producing primordia, yet these growth tips consist of only a few cells and lack the tunica–corpus structure (Figure 5A). This finding demonstrates that even within angiosperms, the tunica–corpus structure is not indispensable for the functional role of growth tips.

In widely studied model plants like *Arabidopsis*, the tunica–corpus structure’s complexity may have evolutionary significance, which complicates efforts to decouple the roles of genes in structure and function. In contrast, *Wolffia*’s highly simplified growth tips provide an exceptional natural system for studying these processes. By eliminating the confounding influence of complex multicellular structures, *Wolffia* creates new opportunities to precisely investigate the mechanisms that enable growth tips to continuously produce primordia. This breakthrough has significant implications for understanding the functional independence of SAM structures.

### 4.2. Induction of Meristos: Small Molecule Gradients and Mechanical Stress

Tip growth is widely recognized as a defining feature of plant morphogenesis, yet the origin of growth tips remains an open question. While all diploid multicellular structures in land plants derive from the division and differentiation of a zygote, does a zygote itself qualify as a growth tip? The most likely answer is no, which leads to the critical question: how do growth tips originate?

Current knowledge identifies at least three major pathways for the generation of growth tips: (1) directly from cell clusters following zygotic cell division, (2) from the axil of a leaf primordium, and (3) through adventitious bud formation in other locations, as observed in shoot regeneration via tissue culture. These pathways suggest that growth tips are not formed directly from zygotic cell division but rather arise from cells that have undergone partial differentiation.

This implies that the formation of growth tips involves a transitional process, whereby cells with some degree of differentiation revert or progress to a state of “meristematicness”. These transitions are pivotal to understanding how growth tips, regardless of their origin or cell composition, are established in plants.

Before delving into the mechanisms underlying the transition to meristematic growth tips, it is worth exploring the etymology of the word “meristem”. Initially, before I was assigned to teach a course on plant developmental biology for graduate students at the College of Life Sciences, PKU, I assumed “meristem” was a combination of “meri” and “stem”. To confirm this assumption, I checked the dictionary and was surprised to find that the term derives from two Greek components: *meristos* (divided, distributed) and *-em* (denoting the result of an action). Together, “meristem” literally means “dividing cells” (www.dictionary.com).

This etymological understanding aligns with the biological concept of a meristem as a region of actively dividing cells. Given that growth tips are essentially groups (or as mentioned above in bryophytes, a single cell) of dividing cells, the central question becomes, how do these transitions from partially differentiated cells to meristematic states occur?

The most compelling insights into this process come from the work of Xian-Sheng Zhang’s lab at Shandong Agricultural University. They demonstrated that the formation of growth tips begins with a symmetry-breaking event in the auxin distribution pattern at the surface of callus tissues. This initiates dynamic changes in cell fate, mediated by the interaction of several known genes, and ultimately results in the formation of a new growth tip with a tunica–corpus structure [35]. Their findings highlight that the redistribution of small molecules, such as auxin, is the critical first step in transitioning cell status from differentiated to meristematic. Auxin distribution is also essential for the formation of the shoot apical meristem (SAM) during early embryogenesis [36].

Taken together, these studies suggest that growth tip meristematic cells are induced from already partially differentiated cells via small-molecule gradients, with auxin playing an initial role.

If the symmetry-breaking of small-molecule gradients initiates the transition, the next question is, what triggers the symmetry-breaking of these gradients?

Observations of *Wolffia* morphogenesis indicate that mechanical stress may play a key role. After the formation of a leaf primordium, it grows asymmetrically (Figure 4B and Figure 5B). Once the primordium enlarges, its growth tip emerges at the axil of the asymmetrically growing parts (Figure 5B). The earliest observable state (Figure 4B) suggests that geometric “knee points” generated by asymmetric growth may serve as initial sites for growth tip formation.

Mechanical stress is known to significantly influence plant morphogenesis [37]. These findings provide an exciting opportunity to investigate how mechanical forces contribute to the induction of meristematic cells from partially differentiated precursors. Understanding this interplay between mechanical stress and small-molecule gradients could reveal fundamental principles governing plant development.

### 4.3. Shape of Cell Cluster: Foliage and Its Deformation

As mentioned earlier, there are two perspectives on the primary geometric shape of multicellular plant structures: foliage or axial. Geometrically, one-dimensional forms are considered primary. The existence of protonema in bryophytes supports the axial-first hypothesis. In practice, the L-system proposed by Prusinkiewicz and Lindenmayer [10] and the telome theory introduced by Zimmerman [15,16,17] demonstrate how one-dimensional multicellular structures evolve into two-dimensional foliage structures and, ultimately, three-dimensional structures such as sporangia in sporophytes.

If we accept this premise, the deformation of multicellular shapes derived from axial or foliage structures becomes a natural extension. A remarkable example is the *Wolffia* leaf. Initially forming as a small bump, similar to leaf primordia in other plants, it enlarges asymmetrically during growth (Figure 4B). More intriguingly, this asymmetric growth results in a cave-like overall structure through mechanisms that remain unclear. This unique morphogenesis provides a fascinating avenue for investigating how the shape of a cell cluster is determined.

This discussion inevitably leads to the nomenclature of *Wolffia* plantlets. Traditionally, researchers have referred to these structures as “fronds” [20]. According to www.dictionary.com, a “frond” in botany refers to the following:A large, finely divided leaf, typically of ferns or palms;A leaf-like structure not differentiated into stem and foliage, as in lichens.

However, observations of *Wolffia* plantlets reveal clear differentiation. Within the “cave” of the plantlet, a distinct leaf develops alongside a growth tip (Figure 4). The overall structure remains attached to the parent plantlet, where the leaf primordia originated. After detachment, the new plantlet displays a petiole (from the perspective of the leaf) or a stem-like structure (from the perspective of the plantlet), marked by an abscission layer (Figure 6). These observations suggest that referring to *Wolffia* plantlets as “fronds” is no longer accurate.

Since *Wolffia* plantlets continuously generate leaf primordia, which in turn produce growth tips and branches [26], the plantlets meet the criteria for being classified as plants. Given their branch-detaching behavior and minimal size, to distinguish *Wolffia* from other ordinary plants, it is more appropriate to call these structures “plantlets” rather than “fronds”. Furthermore, while *Wolffia* plantlets can be viewed as colonies—each containing several branches enclosed within the “cave” leaf—their simplicity and ability to detach individual branches make them approximate representations of a plant developmental unit (PDU). This characteristic makes *Wolffia* an ideal system for studying the regulatory mechanisms underlying core morphogenetic processes in angiosperms.

### 4.4. Induction: The First Principle in Plant Morphogenesis?

*Wolffia* lacks roots ([26] and references therein). Genomic and transcriptomic analyses of *Wolffia* reveal no loss of known genes essential for root development [26]. Combined with findings from *Arabidopsis* root development studies [38], the limited spatial constraints of *Wolffia*’s growth tip, and the absence of vascular bundles in its plantlets, it is reasonable to hypothesize that *Wolffia*’s rootless phenotype results from the loss of a stable auxin gradient.

Previous studies demonstrated that diploid germ cells in *Arabidopsis* stamens are induced through the interaction of an auxin gradient with the spatiotemporal expression of the *SPL/NZZ* gene [39,40,41]. Additionally, researchers have shown that cambium formation is induced by the asymmetric distribution of small molecules, including auxin, cytokinin, and small peptides [42,43]. Combined with the induction of growth tips discussed earlier, these findings suggest that in many critical morphogenetic events—ranging from the formation of meristematic cells and germ cells to vascular tissues and root structure—the initial trigger is a change in the distribution pattern of small molecules, followed by subsequent changes in gene expression.

Based on these findings, an idea arises that induction may represent a fundamental principle in plant morphogenesis. If true, understanding how small molecules distribute within plant structures could become a highly productive area of research. From this perspective, the simplified structure of the *Wolffia* plantlet offers an ideal experimental system for studying plant developmental mechanisms, analogous to *Caenorhabditis elegans* in animal developmental studies.

### 4.5. Flowering: What Does It Exactly Refer to, and an Opportunity to Address Darwin’s “Abominable Mystery”?

My PhD research focused on the mechanism underlying photoperiod-sensitive male sterility in rice. Since male sterility in this system is sensitive to photoperiod, much like photoperiod-sensitive flowering, I reviewed the literature on flowering extensively. In the late 1980s, flowering studies were dominated by physiological approaches, as genetic methodologies like mutant screening and gene cloning were not yet widespread. In their seminal work *The Physiology of Flowering*, Bernier et al. (1981) proposed the concept of the “flowering syndrome” to define what “flowering” entails [44]. Yet, as I read further, a perplexing question emerged: when flowering is described as a transition from vegetative to reproductive phases, what does “reproductive” truly mean?

The ambiguity arises because many structures sequentially develop during flowering, including petals, sepals, receptacles, stalks, bracts, cauline leaves (in species like *Arabidopsis*), and stems (especially in plants with inflorescences), before the emergence of stamens and ovules. Should all these structures, other than stamens and ovules, be classified as “reproductive organs”? If not, how can flowering be accurately described as a transition to the reproductive phase?

Further complicating matters, studies on flower induction have largely focused on the initiation of inflorescence development over the past decades. In annual plants like *Arabidopsis* and rice, floral organ formation often follows inflorescence initiation relatively quickly and is often treated as a spontaneously occurring process for technical convenience. However, in many perennial plants, such as apple, peach, and pear, floral organ initiation does not happen promptly. Instead, it may extend over several months during summer and fall, with meiosis and gametophyte development occurring only in the following spring after winter dormancy. In some species, like persimmon, floral development halts after sepal primordia formation during summer and resumes the following spring to complete the development of the remaining organs [45,46]. It is implausible that these sequential events proceed unaffected by internal and external changes over such long periods, with seasonal changes, suggesting that multiple inductive mechanisms are likely involved.

In my own work on photoperiodic effects on male sterility, I found that day length affected not only male fertility in the mutant line but also the morphogenesis of other structures, including leaf shape and awn length, in both mutant and wild-type lines [47,48]. Should these effects also be classified as part of “flower induction”? This raises a critical question: is “flower induction” a single event encompassing both inflorescence initiation and all subsequent morphogenetic processes? Or is it merely a trigger for inflorescence initiation, with later stages governed by separate, currently unknown inductive mechanisms?

These considerations highlight the complexity and nuances of the flowering process, challenging simplistic definitions. Understanding the distinct mechanisms driving different stages of floral development offers an opportunity to revisit Darwin’s “abominable mystery”—the origin and rapid diversification of flowering plants—by dissecting the modular, multifaceted nature of flowering as a key evolutionary innovation.

According to Figure 1 and what was discussed in reference [41], it is evident that only stamens and ovules are diploid reproductive organs, as they produce meiotic cells. Given that *Wolffia* produces only stamens and pistils upon “flower induction”, it offers a uniquely simplified system for studying the mechanisms underlying this process. This simplicity allows researchers to focus directly on the core issues without interference from the complexity of intermediate structures typically seen during the transition from leaf primordia to stamens and pistils. Is this not an exciting scenario to explore?

In addition to clarifying the concept of “flower induction”, what we observe in *Wolffia* flower development sheds light on the origin of angiosperms.

Angiosperms are humanity’s most vital plant resource, which is why much of plant biological research focuses on them. However, the origin of angiosperms—when and how they arose—remains unresolved [49].

The structural complexity of flowers may have prevented Darwin from proposing a straightforward explanation for the origin of angiosperms. Since Linnaeus used “sexual organs” as the basis for his plant classification system, defining these organs and explaining the increasing structural complexity and diversity associated with them became critical to understanding angiosperm origins. Several classification systems, such as the Engler and Hutchinson systems, were later derived from Linnaeus, but their differences added further layers of complexity. Combined with ongoing debates about the essential structure of plant bodies (e.g., foliage versus axial origin), this multidimensional controversy has only deepened the mystery.

If the mystery of the origin of angiosperms is rooted in the complexity of flower-related structures, plants with minimal structural complexity might provide a clearer perspective.

In *Wolffia* morphogenesis, only one stamen and one pistil are produced on average (Figure 7; for a detailed morphological description, see reference [26]). By adopting the “axial tree” model or the telome theory, in which the growth tip terminates as a sporangium for germ cell production [10,16,17], and treating stamens as clusters of microsporangia, pistils as derived megasporangia [50]—collectively referred to as modified sporangia—*Wolffia*’s flower primordia can be interpreted as separate growth axes terminating in modified sporangia. From this perspective, the flower structure is significantly simplified: it represents a complex of two axes terminating in heterosporangia—one a cluster of microsporangia (stamen) and the other a derived megasporangium (pistil)—each independently initiated as branching axes.

Since *Wolffia* is undoubtedly an angiosperm, as confirmed by genomic analyses [26], its morphology suggests that the origin of flowers, and thus angiosperms, was not a single evolutionary innovation but rather the integration of at least three independent evolutionary events:
Modification of sporangia: Microsporangia evolved into anthers (as reviewed by Bai [41]), while megasporangia were enveloped by surrounding structures to form ovules [15,50].Integration of axes: Independent axes terminating in modified microsporangia and megasporangia were combined into a single reproductive structure.Carpel integration with ovules: The carpel, uniquely adapted to collect pollen, increased the efficiency of heterogamete interaction. Contrary to traditional interpretations, this adaptation primarily facilitated gamete encounters rather than providing additional protection for gametophytes [1,50,51].

These insights point to a promising approach to addressing Darwin’s “abominable mystery”. By using *Wolffia* as an experimental system to decipher the mechanisms underlying the three evolutionary innovations, researchers can then apply these findings to investigate how these events integrated into basal angiosperm species, as well as the diversified ways of integration that may be responsible for the rapid diversification of angiosperms. *Wolffia*’s simplicity provides an unparalleled opportunity to dissect these processes and advance our understanding of angiosperm origins.

## 5. Perspectives: A Fascinating Blend of Paradigm Shift in the Interpretation of Plant Morphogenesis and the Emergence of the Second Agricultural Revolution

Throughout my academic career, I have often been asked, “What is the use of your research?” The more I delve into the complexity of plant life, the less confident I am in claiming immediate utility for my studies on plant morphogenesis, whether in specific events or fundamental principles. Regarding *Wolffia*, it is unsurprising that funding agencies have shown little interest, given the plantlet’s minuscule size and apparent lack of application potential. This compelled me to rely on crowdfunding from trusted friends who believed in my vision for *Wolffia* research.

Fortunately, their trust and efforts have been rewarded. Observing *Wolffia* morphogenesis has not only uncovered core processes of plant development, enabling us to address fundamental questions, but also demonstrated the feasibility of dissecting these mechanisms using cutting-edge technologies. Uniquely, the Plant-on-Chip culture system provides an unparalleled platform for exploring previously inaccessible problems, such as studying the behavior of an entire plantlet.

Traditionally, plant scientists have to work with colonies, mixtures of unsynchronized PDUs (plant developmental units). Now, with *Wolffia* plantlets, which approximate PDUs, researchers can analyze detailed morphogenetic events with unprecedented precision. This capability signifies a paradigm shift in how plant morphogenesis is studied.

Since Da-Ming introduced *Wolffia* to me in 1994, I had never considered its application potential—nor that of its close relatives, collectively called duckweed—until I attended a symposium in Chengdu in 2023.

This symposium, co-organized by Hai Zhao of the Chengdu Institute of Biology, CAS, and Zheng-Biao Gu of Jiangnan University, provided a transformative perspective. Their presentations revealed that while duckweed starch (from *Landoltia*, a relative of *Wolffia*) is equivalent in quality to corn starch, its annual yield surpasses corn by 6–8 times on average, significantly improved from previous reports [52,53,54]. Detailed information regarding the measurement and cultural conditions will soon be published in [55].

Corn is widely recognized as one of the crops with the highest photosynthetic efficiency. How, then, can the much smaller duckweeds (even *Landoltia*, larger than *Wolffia*) achieve such dramatically higher yields?

Based on the concept of PDU, I uncovered the secret behind the high productivity of duckweeds, which far exceeds that of corn: the advantage of minimizing the physical distance between the three essential components for photosynthesis—light, water, and CO_2_.

As photoautotrophic multicellular eukaryotes, plants inherently increase their photosynthetic area through continuous growth driven by positive feedback, a fundamental characteristic of living systems [1]. However, for terrestrial plants, this growth creates a side effect: an increasing physical separation between the three essential photosynthetic components. Light comes from the sun above, water comes from the earth below, and CO_2_ is absorbed from the atmosphere. Energy is, therefore, required to overcome this separation, as photosynthesis can only occur when all three components converge at the photosynthetic membranes within chloroplasts.

Duckweeds circumvent this limitation through their aquatic habitat, which naturally minimizes the physical distance between light, water, and CO_2_. Additionally, their ability to detach branches ensures that clones, driven by positive feedback, expand across the water surface. This prevents any significant increase in the distance between the three components. Under optimal conditions, duckweeds maintain continuous annual growth through branching, accumulating starch and other bio-products throughout most of their plant body for easy collection. From this perspective, the superior photosynthetic efficiency of duckweeds as bioreactors is primarily physical rather than biological.

Another significant advantage of using duckweeds as bioreactors is also physical. One major challenge in increasing the yield of terrestrial crops is external stress, whether biotic or abiotic. The high specific heat capacity of water provides a natural buffering system against abiotic stresses, such as temperature fluctuations, for duckweeds. Meanwhile, the simpler microbial communities in aquatic environments compared to soil [56] greatly reduce the biotic stress faced by duckweeds, further enhancing their efficiency compared to terrestrial crops.

It is truly fascinating that efforts to find a suitable experimental system for investigating the regulatory mechanisms underlying plant morphogenesis and the reinterpretation of plants as colonies of unsynchronized PDUs have serendipitously aligned with a significant application potential rooted in the clonal growth habit of duckweeds!

As the world faces the challenge of feeding an ever-accelerating population, various strategies have been proposed [57]. However, most of these focus on improving existing terrestrial crops that have already been domesticated over thousands of years. Given the indispensable grain-to-straw ratio and the inherent physical limitations posed by the separation of the three essential photosynthetic components in these crops, their potential for future improvement is marginal. In this context, the unique “Two Minima” characteristics of duckweeds—minimizing the physical distance for photosynthesis and minimizing stress exposure—open an entirely new horizon for utilizing aquatic plants as bioreactors for food production. If the domestication of terrestrial plants marked the first agricultural revolution, the domestication of aquatic plants like duckweeds could herald a second agricultural revolution, shifting agriculture from terrestrial to aquatic.

Despite duckweeds’ potential, their study has received astonishingly little investment compared to terrestrial crops. While their starch yield surpasses corn by 6–8 times, research funding for duckweeds remains negligible. This imbalance, however, represents an opportunity. The issues outlined above are likely just the tip of the iceberg. Beyond the horizon lies an extraordinary realm of scientific discovery and investment potential. What is critical now is a paradigm shift in our conceptual framework. Just as we have reimagined plants as coral-like colonies rather than unitary individuals, we must shift our focus from further manipulation of terrestrial grain crops to the domestication and utilization of aquatic plants like duckweeds. This shift has the potential to revolutionize agriculture and address global food security in ways previously unimagined.

## Figures and Tables

**Figure 1 plants-14-00396-f001:**
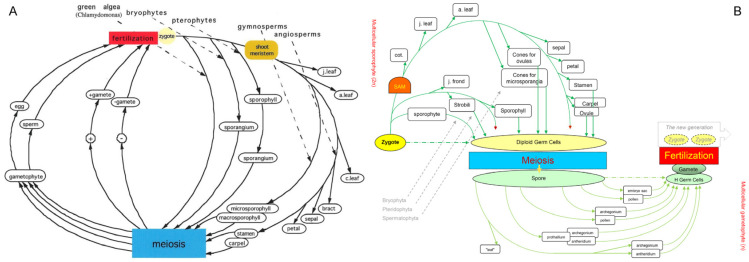
Diagram comparing life cycles or alternation of generations in three plant phyla. (**A**) A version from 1993, using meiosis, fertilization, and zygote as reference points common to eukaryotes. (**B**) Elaboration levels around the core processes in the life cycles of three plant phyla: Bryophyta, Pteridophyta, and Spermatophyta. The sexual reproduction cycle [4], spanning from one zygote to the next generation’s zygote through meiosis and fertilization, serves as the backbone of life cycles across these groups. Green arrows represent organ differentiation in the diploid phase, light green arrows represent differentiation in the haploid phase, and dark red arrowheads indicate unlimited tip growth. The gray arrows labeled Bryophyte, Pteridophyta, and Spermatophyta indicate the representative morphogenetic processes occurring in the diploid phase of the three plant phyla, respectively. For the haploid phase, the first and second lines from left to right represent morphogenetic processes in Bryophyta, the third and fourth, Pteridophyta, and the fifth and sixth, Spermatophyta. Abbreviations: cot., cotyledons; j. leaf, juvenile leaf (e.g., rosette leaves in *Arabidopsis*); a. leaf, adult leaf (e.g., cauline leaves in *Arabidopsis*). Reprinted from [8].

**Figure 2 plants-14-00396-f002:**
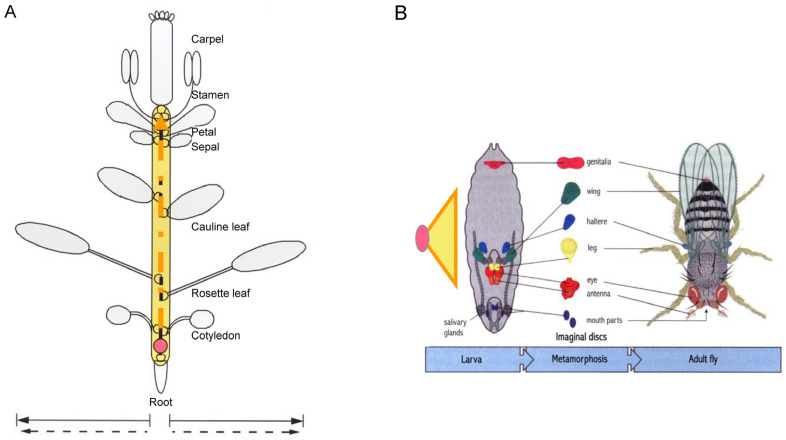
Comparison of developmental units in plants and animals required for life-cycle completion. In plants, a growth tip derived from a zygote (pink circle) can produce numerous lateral organs and branches, with seven organ types completing the life cycle in *Arabidopsis* (**A**). Half circles along the dashed orange arrow represent organ primordia. In animals, the fundamental structure required for life-cycle completion is the embryo, elaborated from zygote to larva (**B**). Embryogenesis is represented by an orange-lined yellow triangle. Unlike animals, which possess a finite number of organs in a fixed pattern, plants rely on the imaginal unit shown in (**A**), referred to as a “developmental unit”, rather than the whole plant. The equivalent of an animal embryo is depicted as the yellow area in (**A**), with dashed orange lines indicating an open but ultimately limited process. (**B**) Adapted from Wolpert et al., 2007, *Principles of Development*, and [8].

**Figure 3 plants-14-00396-f003:**
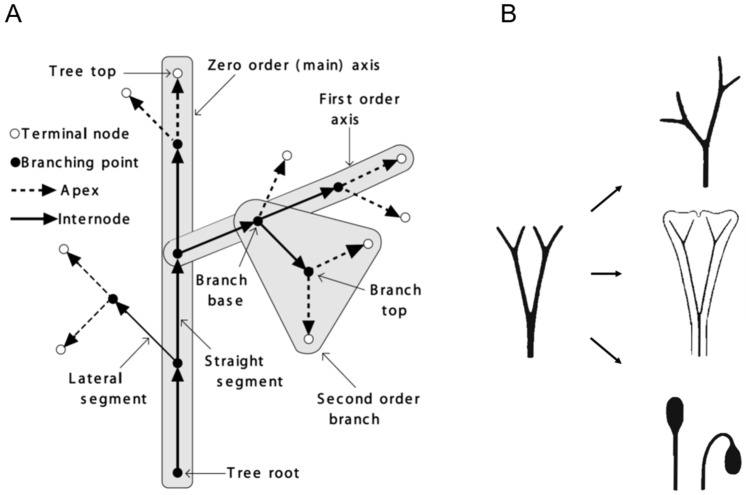
Diagrams of the axial tree model and the telome theory. (**A**) The axial tree concept from [10] (www.algorithmicbotany.org). (**B**) The telome theory posits that plant bodies originated from branched axial structures, with branches evolving into asymmetric axes (**right upper**), webbed foliage (leaves, **right middle**), and terminal sporangia (e.g., capsules in bryophytes, sporangia in pteridophytes, stamens and ovules in angiosperms, **right lower**). Adapted from [15].

**Figure 4 plants-14-00396-f004:**
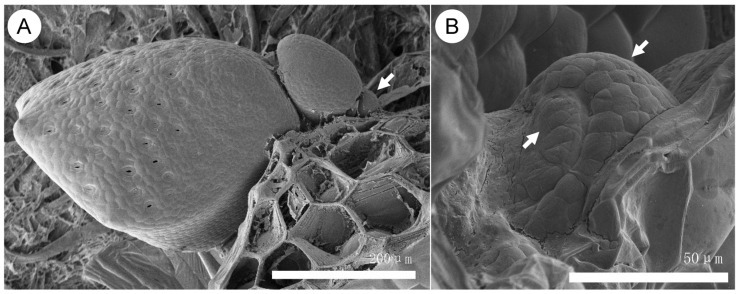
Sequential initiation of leaf primordia and asymmetric growth of a leaf primordium. (**A**) Sequential initiation of leaf primordia, from largest (left) to smallest (arrow pointed). (**B**) Asymmetric growth of a leaf primordium, with a fast-growing region (upper-right arrow) and a slow-growing region (lower-left arrow). For more detailed morphological descriptions, refer to [26].

**Figure 5 plants-14-00396-f005:**
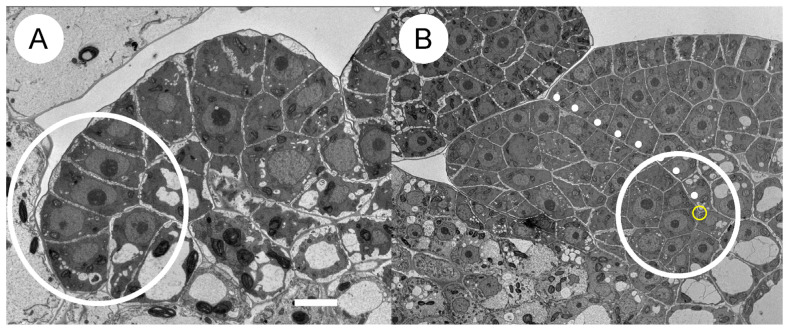
Longitudinal sections of growth tips and leaf primordia in *Wolffia* plantlets. (**A**) A longitudinal section of a growth tip in a *Wolffia* plantlet, with a circle highlighting cells with large nuclei and dense cytoplasm, likely including growth tip cell(s). (**B**) A longitudinal section of a leaf primordium, showing a dotted line indicating the border between fast- and slow-growing regions. The circled junction marks where a new growth tip may form, allowing the leaf primordium to branch. See [26] for details. Bar: 50 µm.

**Figure 6 plants-14-00396-f006:**
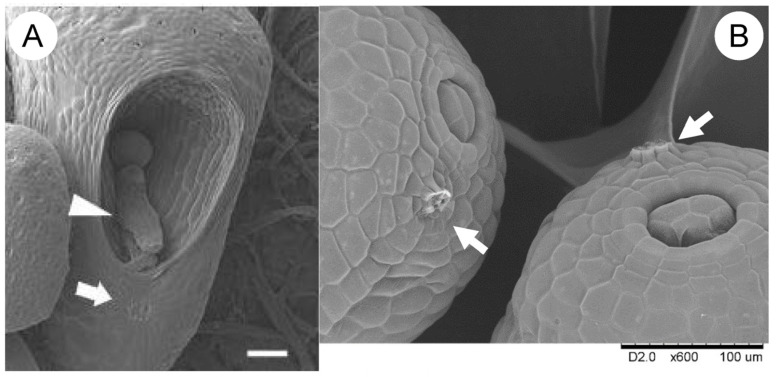
Petiole or stem-like structures and abscised scars in *Wolffia* plantlets. (**A**) An outside-in view of a *Wolffia australiana* plantlet. The arrowhead indicates the petiole (before forming a new growth tip) or stem-like structure (after growth tip formation). The arrow shows an abscised scar. For a detailed morphological description, see [26]. (**B**) Abscised scars (arrows) in *Wolffia globosa* plantlets, demonstrating this feature across the genus. Bar (**A**): 100 µm.

**Figure 7 plants-14-00396-f007:**
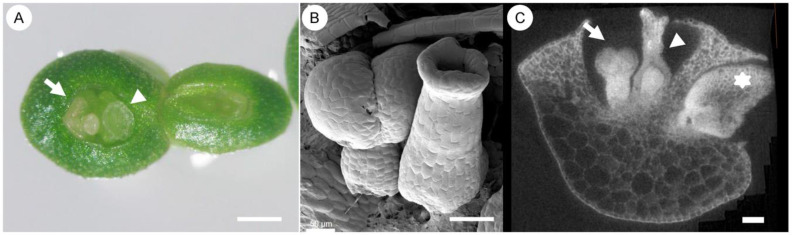
Flowers and floral organs in *Wolffia* plantlets. (**A**) Two flowering plantlets with a stamen (arrowhead) and a stigma (arrow) visible in surface cracks under a dissecting microscope. (**B**) A stamen (**left**) and gynoecium (**right**) inside a plantlet. (**C**) CT imaging of a plantlet showing a gynoecium (arrow), a stamen (arrowhead), and a branch (star) that may develop into a new plantlet. Bar (**A**): 200 µm; (**B**,**C**): 100 µm. For a detailed morphological description, see reference [26].

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
