# Peer review of "On the Plant Developmental Unit: From Virtual Concept to Visual Plantlet"

_plants, 2025, doi:10.3390/plants14030396_

Round 1
Reviewer 1 Report
Comments and Suggestions for Authors
The main purpose of the paper is to represents the potential of Wolffia as a model plant in developmental biology research. Using a revitalization of older hypotheses and concepts of developmental biology, the author highlights the shortcomings of modern and widely accepted concepts and model organism.
At the beginning of the article, the author points out contradictions in the main concepts of the development of flowering plants. First contradictions are about the flower and differences in definition what flower really is. The complexity of flower of arabidopsis or maize leads to problems in research and interpretation and often in detection of relevant developmental/reproductive and also evolutionary events.
The next contradiction is plant developmental determinacy or indeterminacy. In this part I disagree with the fact that the alternation of generations is necessarily opposed to indeterminacy, especially if we consider that the flower meristem, flower organs and flower itself are determined. Cited literature here is mostly in Chinese, so I miss additional interpretations.
The third topic is about the default state in plant development, is it vegetative which carries out the photosynthesis, or reproductive where the vegetative phase is considered only as a link between the zygote and meiosis.... An interesting topic that raises the importance of Wolffia.
After that, the term PDU and the idea of ​​the development of plants as coral-like structures consisting of repeating PDUs are introduced. Looking at the comparison in Figure 2, we certainly meet an interesting approach and the possibility of bypassing the complexity of the plant body, with the aim of answering essential questions of developmental biology. Wolffia, based on such interpretation, would really be a good model organism. The term of a virtual embryo is interesting and opens up a dilemma about the significance of the seed as something that terminates embryogenesis.
Here, I suggest interpretation of evolutionary aspects including evolutionary appernace of sexual embryogenesis, different types of asexual embryogenesis and seed. Does Wolffia produces somatic embryos, how does Wolffia “behave” on these issues?
The main shortcomings of the work that have to be corrected or omproved are listed below:
The large presence of literature in Chinese. In this regard, the text needs additional interpretation (mentioned earlier)
Figure 4 and figure 5 – the descriptions of the images are replaced in the version that I have
Pg 8 lines 28 and 29 - I miss the interpretation of functional zones of SAM (Organizing center, stem cells etc.) Tunica and corpus are not of strong functional importance.
Pg 9 lines 34 - The paragraph is very speculative, include the role of cytokinins for establishment of shoot meristamatic zone, auxin here is player No.2. Auxins are more important for root establishment and leaf induction.
I would also suggest a paragraph about current knowledge on developmental genetics in Wolffia.
Despite this, I considered this viewpoint valuable and interesting and I suggest acceptance for publication after minor revisions.
Reviewer 2 Report
Comments and Suggestions for Authors
Review of the article
On Plant Developmental Unit: From a Virtual Concept to a Visual Plantlet
Summary
This viewpoint article proposes Wolffia as a model system for plant development. After a short introduction the article is divided into four sections: 1. Examples of Self-Contradictory Notions in Plant Biology; 2. The Concept of PDU, Rediscovery of Wolffia, and the Crowdfund Project; 3. Fundamental Issues in Plant Morphogenesis Revealed by the Wolffia Investigation and the PDU Concept, and 4. Perspectives: A Fascinating Blend of Paradigm Shift in Interpretation of Plant Morphogenesis and Emergence of Second Agricultural Revolution
General comment
The interest of the diverse sections is varied, for example some of the examples mentioned in section 1 are not so contradictory. The flower is a reproductive organ but can also be interpreted as a compressed shoot, plant development is indeterminate in comparison with animals, but it is not absolutely indeterminate, and so on…
The following sections of this review address some aspects of the work that can be improved. The first mention of Wolffia must include the botanical author’s name (Wolffia Horkel ex Schleid), and would be informative to include also the family name: Wolffia Horkel ex Schleid (Araceae).
Comments by sections
Introductory sentences: The way to present the work is correct. Probably it could be interesting to add here some more data relative to the Biology of Wolffia (See last 2 paragraphs of comments to sect. 4 below). In addition, two points need your attention:
Line 23: Humans, on the other hand, have historically paid little attention to what plants are 23 or how they live, focusing primarily on which parts are edible for millennia.
Plants have many utilities apart from being edible. They provide drugs, medicaments and cosmetics, construction materials for buildings, ships, furniture, shelter, clothing etc… Additionally one of the main aspects of plants, consists in their similarities with other objects including geometrical and astronomical forms such as the sunflower [Helianthus annuus L. (Asteraceae)], or the genus Stellaria L, reminding the presence of an order in Nature.
Line 27: Please think about the following sentence:
Until humans can intentionally alter the way plants live, these explanations remain irrelevant to the plants themselves
This is an example of personification and should be avoided. In fact, we don’t know what is or can be relevant or irrelevant to plants themselves and this is not the subject of research in Plant Biology. Nevertheless, it is doubtful that alterations made by humans would be beneficial for plants (neither for humans).
1. Examples of Self-Contradictory Notions in Plant Biology
1.1. Is a Flower a Reproductive Organ or a Compressed Shoot?
There is no contradiction in these notions. A flower may be perfectly both a Reproductive Organ and a Compressed Shoot at the same time.
1.2. Is the Plant Developmental Program Indeterminate or Determinate?
It is indeterminate in comparison with animals, but it is not absolutely indeterminate. You could introduce the concept of “open limited program” here.
1.3. Is the Default State in the Plant Life Cycle Reproductive or Vegetative?
There is no need for a default state. The fact that a mutation blocks rosette development in Arabidopsis is not proof enough for the demonstration.
1.4. Is a Plant a Unitary-Animal-like Individual or a Coral-like Colony?
1.5. Is a Plant Body Derived from a Foliage or Axial Structure?
Line 190:
These two perspectives highlight a fundamental geometric distinction: axial structures are one-dimensional, while foliage structures are two-dimensional.
This is an exaggerated model. In nature there are not unidimensional structures. Even the smallest cell is tridimensional.
Please check legend to Figure 5 A legend corresponds to B in Figure and B legend corresponds to A in Figure.
2. The Concept of PDU, Rediscovery of Wolffia, and the Crowdfund Project
Serious problems arise in the legend to Figure 4. Please pay attention to this important issue because most probably Figures 4 and 5 are changed.
[Figure 4. Sequential initiation of leaf primordium and asymmetric growth of a leaf primordium. A) Sequential initiation of leaf primordia, from largest (left) to smallest (arrow). B) Asymmetric growth of a leaf primordium, with a fast-growing region (upper-right arrow) and a slow-growing region (lower-left arrow). See Li et al., 2023 for detailed morphological descriptions.
1) From largest to smallest what? Cells? Leaves?
2) No arrows are visible on the Figure
3) Morphological descriptions should be included in the figure legend. The reader does not need to go and see any other publication.]
Line 253: seeds have not been observed in lab-grown Wolffia.
Could you please indicate some characteristics of wild grown seeds of Wolfia?
3. Fundamental Issues in Plant Morphogenesis Revealed by the Wolffia Investigation and the PDU Concept
3.1. Tip Growth and Shoot Apical Meristem: Decoupling Structure and Function
Line 281. Your attention please:
This structural diversity suggests that specific multicellular configurations may not be essential for the functional capabilities of tip growth.
Better:
This structural diversity suggests that specific multicellular configurations may not be essential for the functional capabilities of tip growth in all organisms.
Figure 5. See commentary to Figure 4. Correct positions of Figures 4 and 5. Again, it is not appropriate to refer to Figures in other articles (Introduce in this Figure legend the information required instead of mentioning: See Li et al., 2023 for details.)
3.2. Induction of Meristos: Small Molecule Gradients and Mechanical Stress
3.3. Shape of Cell Cluster: Foliage and Its Deformation
Line 363:
As mentioned earlier, there are two perspectives on the primary geometric shape of multicellular plant structures: foliage or axial. Geometrically, one-dimensional forms are considered primary.
But:
As mentioned in my commentary to sect. 1.5. In nature there are not unidimensional structures, nor bi-dimensional structures. The simplest known structure is tridimensional.
Legend to Figure 6. Same as in legend to Figure 4: Descriptions must be in the legend to the figure and not refer to figures in other articles.
3.4. Induction: The First Principle in Plant Morphogenesis?
Line 405: Wolffia lacks roots (Figure 1 in [26] and references therein).
Please avoid quoting figures from other articles. Introduce in your article the required figures or explain their meaning with words.
3.5. Flowering: What Does It Exactly Refers To, and an Opportunity to Address Darwin’s 426 “Abominable Mystery”?
Line 452. Please make clear:
It is implausible that these sequential events proceed unaffected by internal external changes over such long periods, with seasoning changes, suggesting that multiple inductive mechanisms are likely involved.
You mean internal and external changes?
The expression Abominable mystery is mentioned many times in this article, at least six times in this section. I find this expression often in the literature, although it is not particularly informative, interesting, or even correct (the word Abominable is unaccurate here, please see: https://www.merriam-webster.com/thesaurus/abominable). I would suggest reducing their mentions to a number comprised between 0 and 2. Preferably 0 after consulting the Dictionary.
Figure 7: Again: Don’t refer to figures in other articles. Include in this Figure legend all the information required. Add size of bar.
4. Perspectives: A Fascinating Blend of Paradigm Shift in Interpretation of Plant Morphogenesis and Emergence of Second Agricultural Revolution
Much emphasis is put in the agricultural application of Wolffia, in particular in this sentence:
Their presentations revealed that while duckweed starch (from Landoltia, a relative of Wolffia) is equivalent in quality to corn starch, its annual yield surpasses corn by 6–8 times on average [50].
But it refers unfortunately to a paper in the press. Could you give more details about this data?
This chapter on perspectives is focused on agricultural approach. Could you mention some more basic data about the biology of Wolffia, such as for example: Genome size, number of chromosomes, ploidy status. Are there dioecious species or under some conditions can be produced specifically male or female plants?
Also, being a plant so small, how was attributed to the family Araceae and not to any other family? Was it based on DNA sequence analysis?
References
Authors names and abbreviations with corresponding punctuation signs in Plants format.
Journal’s year in Plants format.
Round 2
Reviewer 2 Report
Comments and Suggestions for Authors
Review of the article:
On Plant Developmental Unit: From a Virtual Concept to a Visual Plantlet
Line 7 The way to spell Wolffia the first time it appears is:
Wolffia Horkel ex Schleid
https://powo.science.kew.org/taxon/urn:lsid:ipni.org:names:331309-2
(no parentheses needed). The sentence should remain:
This study introduces the concept of the Plant Developmental Unit (PDU) and validates its application using Wolffia Horkel ex Scheid (Araceae), as a model system for exploring fundamental processes in plant morphogenesis.
Figure 1: Grey arrows point to Bryophyta, Pteridophyta and Spermatophyta above. Could also grey arrows point in these directions below?
Line 425: Change distinghuis to distinguish.
Sect. 4. Please pay attention to this affirmation (lines 578-582):
Their presentations revealed that while duckweed starch (from Landoltia, a relative of Wolffia) is equivalent in quality to corn starch, its annual yield surpasses corn by 6–8 times on average [55], significantly improved to previous reports [56-58].
In so far, the article demonstrates great objectivity and provides well-elaborated data. It is also necessary to accompany these statements with acurate reports of measurements and objective data:
What measurements of starch quality have been carried out?
What yield measurements exist, and under what conditions?
These aspects need to be explained.
References:
14. natureal to natural
Author Response
Review of the article:
On Plant Developmental Unit: From a Virtual Concept to a Visual Plantlet
Line 7 The way to spell Wolffia the first time it appears is:
Wolffia Horkel ex Schleid
https://powo.science.kew.org/taxon/urn:lsid:ipni.org:names:331309-2
(no parentheses needed). The sentence should remain:
This study introduces the concept of the Plant Developmental Unit (PDU) and validates its application using Wolffia Horkel ex Scheid (Araceae), as a model system for exploring fundamental processes in plant morphogenesis.
Thank you! I changed accordingly!
Figure 1: Grey arrows point to Bryophyta, Pteridophyta and Spermatophyta above. Could also grey arrows point in these directions below?
Thank you for your very careful observation! These arrows were added for teaching. As I have used the diagram for many times, I almost neglect their existence and not explain them in the figure legend. I added two sentences to explain them.
Line 425: Change distinghuis to distinguish.
Thank you very much for the sharp observation!
Sect. 4. Please pay attention to this affirmation (lines 578-582):
Their presentations revealed that while duckweed starch (from Landoltia, a relative of Wolffia) is equivalent in quality to corn starch, its annual yield surpasses corn by 6–8 times on average [55], significantly improved to previous reports [56-58].
In so far, the article demonstrates great objectivity and provides well-elaborated data. It is also necessary to accompany these statements with acurate reports of measurements and objective data:
What measurements of starch quality have been carried out?
What yield measurements exist, and under what conditions?
These aspects need to be explained.
Thank you for the concern. I consulted the corresponding author of the manuscript and was informed that the manuscript has been gone through the review process and accepted for publication. Taking the main theme of this article is not regarding to the technical details, not only for the duckweed application, but also for many other experiments with theoretical perspective, I added a sentence to refer the issue you concerned to the reference 55. I hope this would compromise you reasonable concern and the main theme of this article.
References:
- natureal to natural
Thank you very much! I have corrected.
